# COVID-19 Recovery Time and Its Predictors among Hospitalized Patients in Designated Hospitals in the Madhesh Province of Nepal: A Multicentric Study

**DOI:** 10.3390/healthcare12171691

**Published:** 2024-08-24

**Authors:** Jitendra Kumar Singh, Dilaram Acharya, Salila Gautam, Dinesh Neupane, Bishnu Bahadur Bajgain, Raman Mishra, Binod Kumar Yadav, Pradip Chhetri, Kwan Lee, Ankur Shah

**Affiliations:** 1Department of Community Medicine, Janaki Medical College, Tribhuvan University, Janakpur 456000, Nepal; jsingdj@gmail.com; 2Department of Medicine, Cumming School of Medicine, University of Calgary, Calgary, AB T2N 4N1, Canada; dilaram.acharya@ucalgary.ca; 3Independent Researcher, 113 Martinvalley Mews NE, Calgary, AB T3J 4W2, Canada; salilagautamacharya@gmail.com; 4Department of International Health, Bloomberg School of Public Health, Johns Hopkins University, 615 N Wolfe St Suite E8527, Baltimore, MD 21205, USA; dneupan2@jh.edu; 5Department of Paediatrics, Cumming School of Medicine, University of Calgary, Calgary, AB T2N 1N4, Canada; bishnu.bajgain@ucalgary.ca; 6Department of Medicine, Janaki Medical College, Tribhuvan University, Janakpur 456000, Nepal; ramanjnp2006@gmail.com; 7Department of Biochemistry, Madhesh Institute of Health Sciences, Janakpur 456000, Nepal; binod3aug@gmail.com; 8Department of Community Medicine, Universal College of Medical Sciences, Tribhuvan University, Rupandehi, Siddharthanagar 32900, Nepal; chhetriprdp@gmail.com; 9Department of Preventive Medicine, College of Medicine, Dongguk University, Gyeongju 38066, Republic of Korea; 10Department of Radiology, Madhesh Institute of Health Sciences, Janakpur 456000, Nepal; ankurshahatwork@gmail.com

**Keywords:** COVID-19, multicentric study, recovery time, predictors, Nepal, hospital admission

## Abstract

This study aimed to determine COVID-19 recovery time and identify predictors among hospitalized patients in the Dhanusha District of Madhesh Province, Nepal. This hospital-based longitudinal study involved 507 COVID-19 patients admitted to three distinct medical facilities for therapeutic intervention between April and October 2021. Data were collected for patient demography, symptoms, vital signs, oxygen saturation levels, temperatures, heart rates, respiratory rates, blood pressure measurements, and other health-related conditions. Kaplan–Meier survival curves estimated the recovery time, and a Cox proportional hazard model was used to identify the predictors of recovery time. For the total participants, mean age was 51.1 (SD = 14.9) years, 68.0% were males. Of the total patients, 49.5% recovered, and 16.8% died. The median for patient recovery was 26 days (95% CI: 25.1–26.7). Patients with severe or critical conditions were less likely to recover compared to those with milder conditions (hazard ratio (HR) = 0.34, 95% CI: 0.15–0.79; *p* = 0.012). In addition, an increase in oxygen saturation was associated with an elevated likelihood of recovery (HR = 1.09, 95% CI = 1.01–1.17, *p* = 0.018). This study underscores the need for early admission to hospital and emphasizes the targeted interventions in severe cases. Additionally, the results highlight the importance of optimizing oxygen levels in COVID-19 patient care.

## 1. Introduction

COVID-19, caused by Severe Acute Respiratory Syndrome Coronavirus Type 2 (SARS-CoV-2), is still a significant global health problem with changing emergence of its new variants such as Omicron BA.2 and BA.5 and the recombinant XBB [1,2]. There is a need for continued further research and innovation in this area. Although COVID-19 vaccination has successfully reduced the global burden of the virus, several important factors need be considered, including concerns about vaccine reliability, fears of potential side effects, and mass media influence, which have led to hesitancy among families and even health professionals, resulting in decreased vaccination coverage [3]. As of 9 August 2023, there had been more than 760 million confirmed cases and 6.9 million deaths worldwide since December 2019 [4]. SARS-CoV-2 primarily affects the human respiratory system, with patients potentially exhibiting a range of manifestations ranging from asymptomatic cases to atypical symptoms like hyposmia, nasal congestion, rhinorrhea, cough, abdominal discomfort, vomiting, and diarrhea, or more classic symptoms such as fever, headache, dry cough, and dyspnea [4,5].The symptoms are categorized into different levels of severity, ranging from mild and moderate to severe and critical illness [6]. Adverse health outcome of SARS-CoV-2 are well documented, impacting poor prognosis of the disease and leading to multi-organ dysfunction, including respiratory failure, septic shock, acute cardiac damage, acute renal failure, psychological illnesses, and death, depending on affected participants’ attributes [7,8,9]. Furthermore, several reports highlight adverse long-term health outcomes, significant healthcare and economic burdens, as well as a diminished quality of life [8,10,11,12,13].

The length of stay (LOS) and recovery time have a substantial impact on the healthcare delivery system. This leads to an increased demand for healthcare service providers, increased pressure on healthcare facilities, burnout among health workers [14,15], and heightened risk of hospital-acquired infections [16,17]. These factors further complicate the related burden on health systems. LOS in hospital and recovery times as a result of SARS-CoV-2 infections vary depending on the various health conditions such as presence of pre-existing morbid conditions, population demographics, personal habits, availability and appropriateness of treatment of patients, and use healthcare facilities among others [18]. A recent systematic review and meta-analysis reported a median hospital LOS ranged from 4 to 53 days within China, and 4 to 21 days outside of China [15]. The same study [15] demonstrated similar distribution of LOS for those who were admitted to intensive care units (ICUs) of the hospitals for treatment (median interquartile range (IQR) of 8 (5–13) days for China and 7 (4–11) days outside of China)).

A number of studies highlighted the predictors of recovery time of hospitalized patients. Abrahim, S. A. et al. (2020) [19] reported that the rate of recovery from SARS-CoV-2 infection was 93% higher in those with at least one comorbidity than in those without any comorbidity and 36% higher in males than in females. However, recovery time was not affected by blood type, body mass index (BMI), and presence of signs or symptoms. Other studies from Ethiopia, Italy, and China reported that critical stage, severe stage, mechanical ventilation, treatment center [18], old age [7,20], female sex [21], and co-morbidities [7,21] were significant predictors of recovery rate among hospitalized patients.

Understanding the length of hospital stay and predictors of recovery time are meaningful to make informed decision by clinicians and other stakeholders. Nepal reported its first case of COVID-19 on 13 January 2020 [22]. As of 10 November 2023, the confirmed cases and deaths were 1,003,450 and 12,031, respectively [23]. In 2020, the Madhesh Province of Nepal accounted for nearly 50% of the total cases and related fatalities in the country [24]. To address the identified gap, this study specifically aims to determine the recovery time and identify its predictors among hospitalized patients in the Madhesh Province of Nepal, thereby providing valuable insights that can inform healthcare strategies and interventions in the region. 

## 2. Materials and Methods 

### 2.1. Study Design and Setting 

We undertook a multicenter retrospective longitudinal study spanning from 9 April 2021 to 20 November 2021. The study was carried out at hospitals specifically designated for COVID-19 cases in the Dhanusha District of the Madhesh Province in Nepal. Data were collected from three distinct medical facilities: a provincial government medical college hospital, Madhesh Institute of Health Sciences, a private medical college, Janaki Medical College Teaching Hospital (affiliated with Tribhuvan University), and a private hospital, Janaki Health Care and Teaching Hospital. All function as tertiary care teaching hospitals, actively engaged in patient care, medical education, and research endeavors. Moreover, they offer advanced medical services to individuals referred from the surrounding four to five districts of the Madhesh Province in Nepal. 

### 2.2. Study Population

In this hospital-centric study, all individuals afflicted with COVID-19 who were admitted to three distinct medical facilities for therapeutic intervention and subsequently subjected to real-time RT-PCR testing to authenticate their infection were included [25]. Incomplete medical data without date of admission, date of discharge, date of demise, and the duration of their hospital stay were excluded. The study captured a comprehensive record of the diverse medical interventions and therapeutic modalities administered to each patient.

### 2.3. Sample Size and Sampling Technique

Out of the aggregate of 566 patients who were admitted to the specified medical facilities, a discernible set of 525 medical records was successfully ascertained from the medical archives of the three hospitals. Subsequently, a cumulative total of 507 medical records effectively met the inclusion criteria for the study (Figure 1). Of 525 subjects, a subset of eighteen medical records displayed insufficiencies, as essential data points, including admission dates, discharge dates, and dates of demise, were found absent in four, ten, and four records, respectively.

### 2.4. Data Extraction and Analysis

Data retrieval covered an exhaustive perusal of the medical records pertaining to each individual admitted under a diagnosis, a time frame spanning from 9 April to 17 October in 2021, with continuous tracking maintained until 20 November in 2021. A thorough retrieval process relied upon the utilization of the hospital’s distinct registration identifier embedded within the medical record repository. We extracted all relevant data from the available records.

The extraction process was executed through the employment of a standardized data extraction form in English. This instrument facilitated the systematic extraction from both charts and records. The domain included patient demography, residence, symptoms, comorbidities, and vital sign metrics, including oxygen saturation levels, temperatures, heart rates, respiratory rates, and blood pressure measurements.

Furthermore, the requisites for supplementary oxygen, mechanical ventilation, and the subsequent status of hospital discharge were also chosen from the records. The outcomes experienced by the patients during their hospital stay, namely recovery, mortality, transfer to a higher-tier medical facility, voluntary discharge, departure against medical counsel (LAMA), or an unresolved outcome status (signifying outcome data unavailable during the data collection phase), were diligently recorded. The recovery duration was defined as the number of days between the initial rRT-PCR positive test for SARS-CoV-2 and two consecutive negative results of the virus obtained through rRT-PCR within a 24 h period. All study participants were admitted to the hospital within 24 h of testing positive results of nasopharyngeal rRT-PCR for SARS-CoV-2.

### 2.5. Data Quality Assurance

Three data collectors, each in their 4th year of medical studies, undertook the responsibility of extracting information from the patients’ records. A comprehensive one-day training session was conducted, thoroughly designed to establish a uniform comprehension of the data collection instrument and the underlying methodology among all participants. The stringent efforts were dedicated to upholding data quality, accomplished through the crafting of a fitting data abstraction mechanism and the implementation of consistent supervision practices. The meticulousness of the process extended to the subsequent phase, where all the collective data underwent a meticulous review by the principal investigator. This review aimed to ascertain the data’s completeness and rationality, further consolidating the overall reliability of the acquired dataset.

### 2.6. Study Variables and Their Measurement

The term “time to recovery” has been precisely delineated as the interval, expressed in days, spanning from the moment of hospital admission for therapeutic intervention to the day of discharge subsequent to the attainment of full restoration. This definition pertains exclusively to those instances recounted within the medical records of patients who were released from hospitalization after a complete recovery, among those who had sought hospital care. Patients whose outcomes encompassed mortality, referral to a higher-tier medical establishment, voluntary discharge at request (DoR), departure against medical advice (LAMA), or cases with an elusive outcome status (attributed to absent outcome data during the data collection phase) were all subject to censorship in the analysis. Patients were stratified into categories of mild, moderate, severe, or critical [26]. We combined the severe and critical categories as their frequency was small. 

### 2.7. Statistical Methods

The cumulative data were entered into Epi Data Entry 3.1 software. Subsequent to this phase, an accuracy assessment was undertaken, coupled with essential edits as necessitated by the context. The data were then transferred into SPSS 23.0 software (SPSS Inc. situated in Chicago, IL, USA) for the analysis. Frequency distribution was used for categorial data. For continuous data, the mean with standard deviation and median with interquartile range (IQR) were presented. Appropriate statistical tools were applied such as independent sample t-test, and Mann–Whitney U test, Chi-square test, and log-rank test. The investigation of survival probabilities over time, both within groups and between them, was estimated by Kaplan–Meier survival curves. A Cox proportional hazard model (Cox PH model) was adopted for estimating recovery time. All variables considered with a *p*-value < 0.2 within the framework of the multivariable Cox model were included. Three models were developed in total. The first model was unadjusted. In the second model (Model-I), we included patient age and hospital-level characteristics such as hospital type, severity at admission, and the need for respiratory support. In the third model (Model-II), we further added patient vital signs, including oxygen saturation, temperature, heart rate, and respiratory rate, alongside the variables from the previous model (Appendix A). The model-building process followed a progressive stepwise approach, and standard measures of goodness of fit, such as the Akaike Information Criterion (AIC) and the Bayesian Information Criterion (BIC), were used to evaluate the models. The foundational assumption of the Cox PH model was rigorously scrutinized via a log(-log) plot, ensuring the uniformity of hazard over time for all pertinent explanatory factors. Furthermore, the assessment of multicollinearity was undertaken by checking the variance inflation factor. The outcomes of this analysis were represented through the estimation of both crude and adjusted hazard ratios (HRs), with 95% confidence intervals (CIs). The association between recovery time and covariates was established at a *p*-value < 0.05.

### 2.8. Ethics Statement

The ethical approval was obtained from the Nepal Health Research Council (reference number 496/2021P). Each participating institution issued a letter of cooperation. De-identified data from the register of healthcare facilities were utilized for analysis by eliminating the need for patient interviews or human involvement. Consequently, the informed consent from human subjects was opted to leave.

## 3. Results

### 3.1. Treatment Outcomes

In a cohort comprising 507 COVID-19 patients, 251 (49.50%) successfully recovered and 85 (16.77%) died. Furthermore, 88 patients (17.36%) voluntarily requested discharge (DOR), 28 (5.52%) were transferred to an advanced medical facility, and the remaining 55 (10.85%) either showed an unknown response to treatment or opted to leave against medical advice (LAMA) (Figure 2). 

Table 1 describes the demographic and clinical profile of the patients. The mean age was 51.09 (SD = 14.92) years. The majority of patients (68.0%) were male, were from Dhanusha district (67.1%), and inhabited urban areas (80.4%). More than half of the patients (56.2%) received treatment in public hospitals, 43.7% had severe or critical illness upon admission, 15.2% of the patients necessitated mechanical ventilation, 18.7% had diabetes mellitus, 10.7% had hypertension, 2.8% had COPD, 0.8% had asthma, 1.2% had chronic cardiac disease (excluding hypertension), 0.8% had tuberculosis, 0.2% had HIV/AIDS, 3.2% had thyroid conditions, and 1.6% had chronic kidney disease in any stage (Table 2). The utilization of respiratory support among the patient cohort was stratified into three distinct categories: no support, employment of an oxygen mask, and reliance on mechanical ventilation. Significant disparities emerged in the statistical analysis pertaining to age, severity upon admission, and the type of respiratory support administered when comparing the groups of individuals who recovered against those who were deceased or were referred to alternative medical facilities for further care (Table 1).

Table 2 shows the patient characteristics based on signs and symptoms reported upon admission, prevailing medical conditions, and vital signs upon their presentation at the hospital. During admission, over 60% of patients reported shortness of breath, fever, and cough. However, the collective signs and symptoms reported at admission did not display statistically significant differences between the patients who eventually recovered (*p* > 0.05).

The vital signs documented upon hospital admission exhibited variations between patients who recovered and those who died or were transferred. Specifically, temperature (*p* = 0.017), oxygen saturation (*p* < 0.0001), heart rate (*p* = 0.039), respiration rate (*p* = 0.018), and diastolic blood pressure (*p* = 0.026) demonstrated significant variations. 

### 3.2. COVID-19 Recovery Time of Patients 

The median duration for patient recovery was estimated at 26 days (95% CI: 25.1–26.7) (Figure 3). Patients admitted to public hospitals exhibited a median recovery time of 9 days (95% CI: 8.05–9.95) compared to 10 days (95% CI: 8.78–11.21) in private hospitals (Figure 4). Patients with mild symptoms showed a median recovery time of 7 days (95% CI: 5.18–8.81), followed by 9 days (95% CI: 7.78–10.22) among patients with moderate symptoms. Patients with severe symptoms experienced a median recovery time of 10 days (95% CI: 8.59–11.40), and patients with critical symptoms experienced a median recovery time of 18 days (95% CI: 11.96–24.03) (Figure 5).

Furthermore, the mode of respiratory support significantly influenced the recovery period. Patients not requiring respiratory support demonstrated the shortest median recovery time of 5 days (95% CI: 4.08–5.91), whereas individuals using an oxygen mask had a longer median recovery time of 10 days (95% CI: 8.98–11.01). Patients necessitating mechanical ventilation displayed a median recovery duration of 22 days (95% confidence interval: 9.13–34.86).

In further analysis, comparing median survival times disclosed notable disparities in recovery duration based on hospital type (*p* = 0.01), severity at admission (*p* < 0.0001), and mode of respiratory support (*p* < 0.0001). However, factors such as area of residence, origin of residence, gender, or age did not significantly affect the recovery time (*p*-value > 0.05) (Table 3).

The median recovery period for patients presenting symptoms upon admission fell within a range of 9 to 10 days. Notably, the presence or absence of fever, congestion, fatigue, shortness of breath, respiratory distress, and headache did not exhibit a significant correlation with the median recovery time (*p* > 0.05). However, diabetes mellitus was notably linked to an extended median recovery time of 11 days (95% CI: 8.07–13.92) compared to patients without diabetes mellitus, who exhibited a median recovery time of 9 days (95% CI: 7.57–10.2) among individuals with pre-existing conditions. This dissimilarity was statistically significant (*p* = 0.025). On the contrary, there was no observed association between the presence of hypertension, chronic obstructive pulmonary disease, asthma, chronic cardiac illness (excluding hypertension), tuberculosis, HIV/AIDS, thyroid disorders, or chronic renal disease and the median recovery time (*p* > 0.05) (refer to Table 4).

In the unadjusted analysis, age, severity upon admission, respiratory support, and oxygen saturation demonstrated significant associations with recovery time (*p* < 0.05). However, in the multivariable analysis (Model-I), age, severity at admission, and oxygen support emerged as significant factors influencing recovery time. For every ten-year increase in age, the risk of recovery decreased by 13% (AHR = 0.87; 95% CI = 0.75–0.95; *p* = 0.006). Patients with moderate, severe, or critical conditions were notably less likely to recover compared to those with milder conditions. The adjusted hazard ratios were 0.54 (95% CI: 0.37–0.80; *p* = 0.002) for moderate and 0.46 (95% CI: 0.29–0.71) for severe/critical conditions. Moreover, patients who received an oxygen mask or mechanical ventilation displayed significantly reduced recovery risks compared to those without respiratory support. The adjusted hazard ratios for oxygen mask were 0.34 (95% CI: 0.24–0.48, *p* < 0.0001) and for mechanical ventilation, 0.11 (95% CI: 0.05–0.51, *p* < 0.0001) (Table 5).

Following adjustments, severity at admission and oxygen saturation emerged as the independent and significant predictors of recovery in the final model (Model-II). Patients with severe or critical conditions were notably less likely to recover compared to those with milder conditions (AHR = 0.34, 95% CI: 0.15–0.79; *p* = 0.012). Moreover, an increase in oxygen saturation was associated with an elevated likelihood of recovery (AHR = 1.09, 95% CI = 1.01–1.17, *p* = 0.018) (Table 5).

## 4. Discussion

This study examined the time to recovery and its predictors among hospitalized patients in Madhesh Province of Nepal. We found the median duration for patient recovery stood at 26 days (95% CI: 25.1–26.7). The patients with severe or critical conditions were less likely to recover compared to those with milder conditions. Furthermore, an increase in oxygen saturation was associated with an elevated likelihood of recovery. 

The recovery time for patients in our study aligns with findings from a study in India and Italy, where the average recovery times were 24 days and 25 days, respectively [27,28], while a subsequent study [29] across eight Indian states revealed varied recovery times ranging from 5 to 36 days, excluding Madhya Pradesh. Notably, Tamil Nadu exhibited the shortest average recovery time at 7 days, followed by Odisha, Karnataka, West Bengal, Kerala, and Chhattisgarh with estimated durations of 13, 17, 11, 14, and 12 days, respectively. Our study revealed a comparatively prolonged median duration for the recovery of hospitalized patients, surpassing findings from similar investigations from Ethiopia [18,19,20]. In these three Ethiopian studies conducted across various regions, the median recovery period for hospitalized patients varied from 10 to 19 days. Furthermore in contrast to our study, studies from the USA [30] and Belgium [31] reported significantly shorter median recovery times, with 7 days and 10–14 days, respectively. The variation in median recovery times among studies on hospitalized patients can be attributed to several factors, including pre-existing health and disease conditions, disease severity, patients’ demographic characteristics, geographic location, healthcare service quality, time to case identification, and early initiation of adequate medical interventions [32]. Several observational studies have indicated that pre-existing conditions and comorbidities can extend the recovery time for hospitalized cases [18,19,33,34]. For instance, Abrahim, S. A. et al. (2020) [19] reported that the rate of recovery was 93% higher in those with at least one comorbidity than in those without any comorbidity. Similarly, SeyedAlinaghi, S. et al. (2021) [34] reported that mild-to-moderate symptoms vs. critical illness or immunocompromised status on hospital admission time to recovery ranged between 10 and 15 days. Pre-existing conditions and co-morbidities can compromise the immune system’s ability to combat COVID-19, consequently leading to an extended time for recovery [35].

Consistent with previous studies from Ethiopia [18,35,36], the USA [37], and China [38], our study revealed that patients with severe or critical conditions were significantly less likely to recover (AHR = 0.34, 95% CI: 0.15–0.79; *p* = 0.012) compared to those with milder conditions. The potential explanation for the poor prognosis of patients admitted with severe or critical conditions may involve the necessity for intensive and sophisticated medical interventions, such as ensuring adequate oxygenation, employing lung-protective ventilation strategies, managing fluids appropriately, administering suitable antibiotics for suspected bacterial co-infections until a specific diagnosis is made, and ensuring the availability of adequate health infrastructure and well-trained healthcare service providers [36,37,38], which are not readily available in the case of developing countries like Nepal. Another contributing factor to the poor recovery of severe/critically ill cases is the reported decrease in platelet, lymphocyte, hemoglobin, eosinophil, and basophil counts, along with an increase in neutrophil count. Additionally, the worsening of the neutrophil/lymphocyte and platelet/lymphocyte ratios has been associated with a deteriorating clinical outcome, consequently prolonging the recovery time [39,40,41]. 

Our study also identified that an increase in oxygen saturation was correlated with an increased likelihood of recovery (AHR = 1.09, 95% CI = 1.01–1.17, *p* = 0.018), aligning with similar observations in several other studies [42,43,44,45]. The interconnection of hypoxia and inflammation at molecular, cellular, and clinical levels [46] implies that acute hypoxemia may heighten neutrophils’ cytotoxic functions, fostering hyperinflammation and consequently prolonging recovery time for patients; therefore, maintaining adequate oxygen saturation in managing hospitalized cases emerges as a potential strategy to alleviate long recovery times and mitigate associated complications and mortality. 

In our multivariable analysis (Model-I), age (increase by 10 years), moderate severity, and the need for respiratory support (oxygen mask and mechanical ventilation) on admission were significant factors affecting recovery time. However, these variables became insignificant after adjusting for age, severity at admission, respiratory support, and oxygen saturation. Elderly hospitalized cases requiring respiratory support on admission are recognized to benefit from intensive healthcare measures, including adequate oxygenation, to reduce prolonged hospitalization and enhance patient survival [18,36,47,48,49].

A key strength of our study lies in the utilization of a secondary dataset derived from three prominent tertiary-level healthcare facilities dedicated to cases in the densely populated Madhesh Province of Nepal. This method ensures a substantial sample size, thereby bolstering the statistical power and external validity of the study for comparable settings. However, it is crucial to interpret the findings cautiously due to the retrospective nature of the data, which only involves one province in Nepal. Generalizations of the study’s findings to all individuals across the country should be restrained. Furthermore, the cross-sectional design of the study limits the researcher from establishing causal associations with the outcome of interest. Finally, we were unable to use electronic data extraction because the investigation information was not available on computer systems, which may have introduced observer bias.

## 5. Conclusions

The median time to recovery for hospitalized COVID-19 patients was 26 days, which is relatively high. The hazard of recovery was higher for those with severe or critical health conditions on admission, and higher oxygen saturation level during treatment increased the likelihood of recovery. Special attention is needed for those patients who are severely/critically ill on admission, as is maintaining an optimum level of oxygen during treatment to reduce the mortality and increase patients’ survival associated with hospitalized COVID-19 patients. These findings could contribute to making informed decisions by healthcare providers and estimating healthcare needs during the COVID-19 pandemic and other similar pandemic crises. Further studies are essential to validate the findings of this study. Additionally, we recommend conducting further research to understand the impact of additional lifestyle-related factors, such as smoking, alcohol consumption, substance abuse, physical activities, and dietary habits, on the length of stay (LOS) and recovery time in hospitalized COVID-19 cases.

## Figures and Tables

**Figure 1 healthcare-12-01691-f001:**
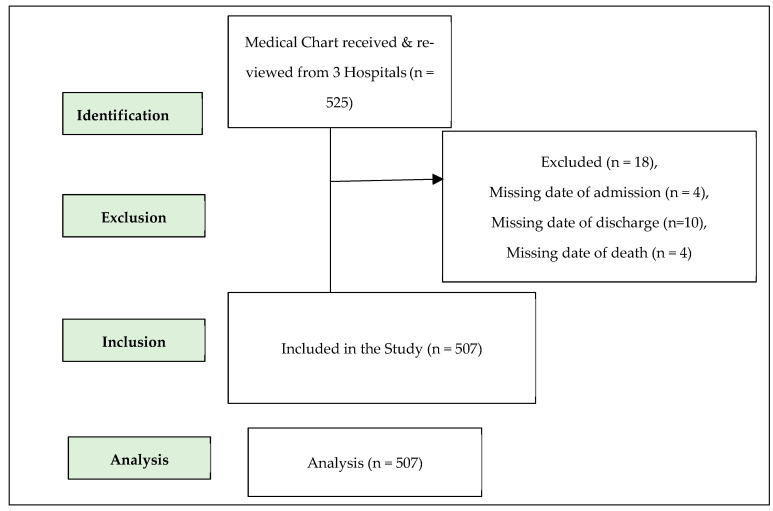
STROBE flowchart for patients’ assessment in three hospitals of Madhesh Province, Nepal.

**Figure 2 healthcare-12-01691-f002:**
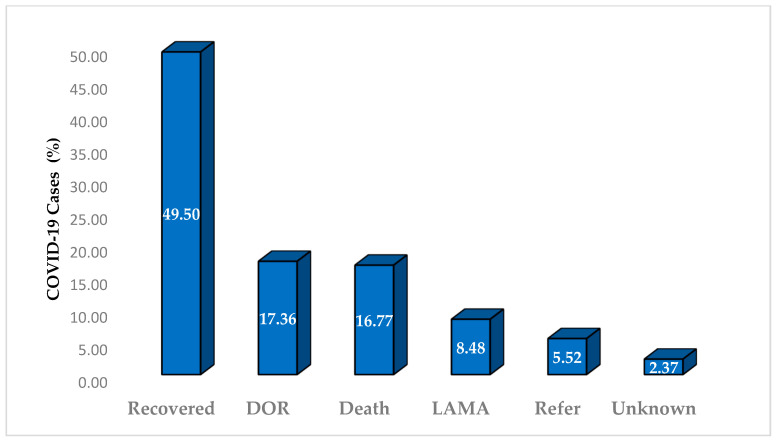
COVID-19 treatment outcomes in designated hospitals (*n* = 507). Abbreviations: DOR, voluntarily requested discharge; LAMA, left against medical advice.

**Figure 3 healthcare-12-01691-f003:**
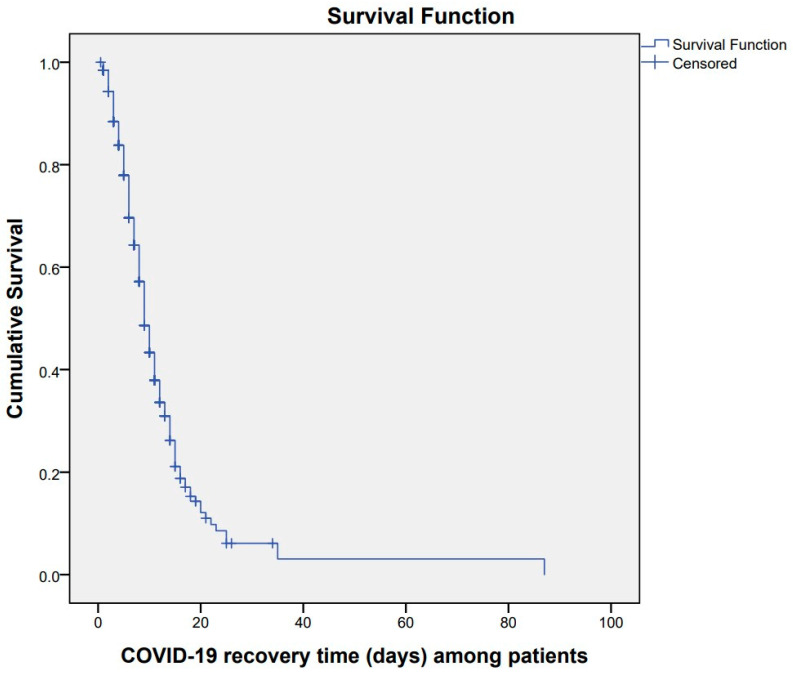
Kaplan–Meier survival estimate of COVID-19 recovery time among patients admitted to designated hospitals, Madhesh Province, Nepal.

**Figure 4 healthcare-12-01691-f004:**
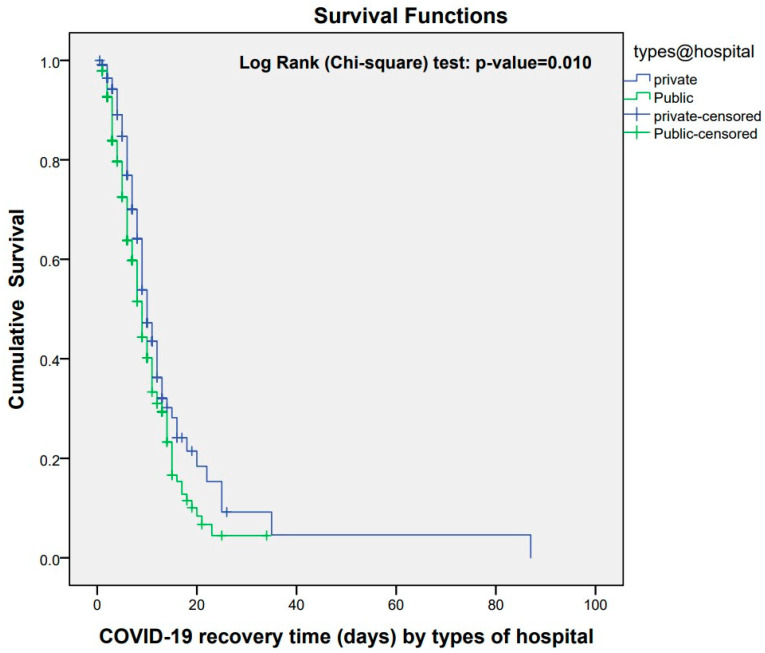
Kaplan–Meier survival estimate for COVID-19 recovery time by types of hospital among patients admitted to designated hospitals, Madhesh Province, Nepal.

**Figure 5 healthcare-12-01691-f005:**
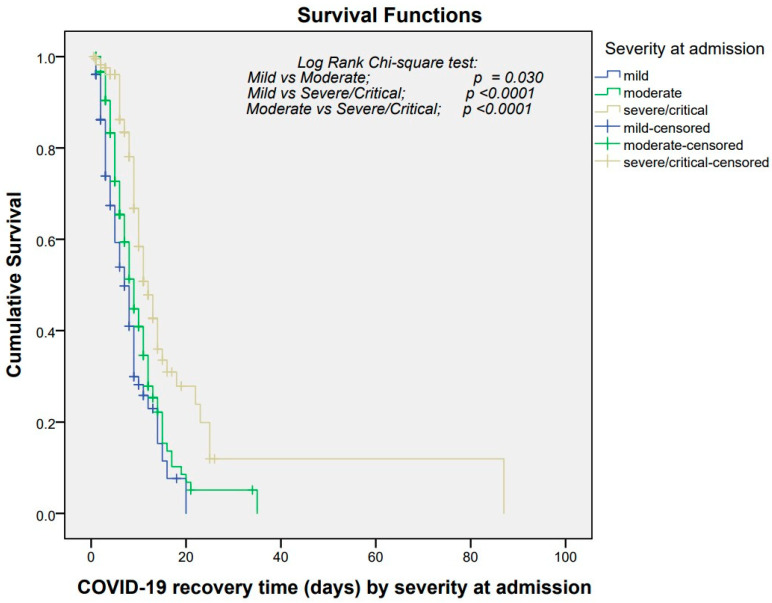
Kaplan–Meier estimate of survival by severity at admission in COVID-19 patients admitted to designated hospitals, Madhesh Province, Nepal.

**Table 1 healthcare-12-01691-t001:** Demographic and clinical characteristics of COVID-19 patients in designated hospitals (*n* = 507).

Variables	All Patients (*n* = 507)	Patients Who Recovered (*n* = 251)	Patients Who Died/Referred * (*n* = 256)	*p*-Value
Age, years				
Mean (SD)	51.09 (14.92)	48.61 (14.99)	53.52 (14.48)	<0.001
Gender				
Male	345 (68.0)	173 (50.1)	172 (49.9)	0.675
Female	162 (32.0)	78 (48.1)	84 (51.9)	
Origin of Residence *				
Dhanusha	255 (67.1)	126 (49.4)	129 (50.6)	0.713
Mahottari	76 (20.0)	36 (47.4)	40 (52.6)	
Sarlahi	26 (6.8)	13 (50.0)	13 (50.0)	
Siraha	17 (4.5)	6 (35.3)	11 (64.7)	
Bara/Parsa/Rautahat/Saptari	6 (1.6)	4 (66.7)	2 (33.3)	
Area of Residence **				
Urban	299 (80.4)	147 (49.2)	152 (50.8)	0.692
Rural	73 (19.6)	34 (46.6)	39 (53.4)	
Types of Hospital				
Public	285 (56.2)	149 (52.3)	136 (47.7)	0.157
Private	222 (43.8)	102 (45.9)	120 (54.1)	
Severity at admission ***				
Mild	103 (22.3)	62 (60.2)	41 (39.8)	<0.0001
Moderate	157 (34.0)	101 (64.3)	56 (35.7)	
Severe	135 (29.2)	51 (37.8)	84 (62.2)	
Critical	67 (14.5)	13 (19.4)	54 (80.6)	
Respiratory support ****				
None	80 (20.0)	62 (77.5)	18 (22.5)	<0.0001
Oxygen mask	260 (64.8)	136 (52.3)	124 (47.7)	
Mechanical Ventilation	61 (15.2)	8 (13.1)	53 (86.9)	

* Missing = 127; ** missing = 135; *** missing = 45; **** missing = 106.

**Table 2 healthcare-12-01691-t002:** Characteristics of COVID-19 patients based on symptoms reported at admission, pre-existing conditions, and vital signs during hospital presentation (*n* = 507).

Variables	All Patients (*n* = 507)	Patients Who Recovered (*n* = 251)	Patients Who Died or Referred * (*n* = 256)	*p*-Value
Symptoms reported at admission				
Shortness of breath	332 (65.5)	156 (47.0)	176 (53.0)	0.233
Fever	310 (61.1)	157 (50.6)	153 (49.4)	0.680
Cough	305 (60.2)	154 (50.5)	151 (49.5)	0.603
Fatigue	56 (11.0)	27 (48.2)	29 (51.8)	0.495
Respiratory distress	16 (3.2)	6 (37.5)	10 (62.5)	0.221
Headache	29 (5.7)	16 (55.2)	13 (44.8)	0.888
Pre-existing conditions				
Diabetes mellitus	95 (18.7)	41 (43.2)	54 (56.8)	0.256
Hypertension	54 (10.7)	27 (50.0)	27 (50.0)	0.686
Chronic obstructive pulmonary disease	14 (2.8)	7 (50.0)	7 (50.0)	0.785
Asthma	4 (0.8)	1 (25.0)	3 (75.0)	0.343
Chronic cardiac disease ‡ (Excluding hypertension)	6 (1.2)	3 (50.0)	3 (50.0)	0.839
TB	4 (0.8)	2 (50.0)	2 (50.0)	0.908
HIV/AIDS	1 (0.2)	0 (0.0)	1 (100.0)	-
Thyroid	16 (3.2)	9 (56.3)	7 (43.7)	0.738
Chronic kidney disease of any stage *	8 (1.6)	2 (25.0)	6 (75.0)	0.250
Vital signs at hospital presentation				
Temperature (°F) [*n* = 328]	98 (97–99)	98 (97–99)	98 (97–99)	0.017
Oxygen saturation (%) [*n* = 475]	94 (88–97)	95 (92–97)	90 (80–95)	<0.0001
Heart rate (beats per min) [*n* = 335]	88 (80–100)	86 (80–97)	89 (80–105)	0.039
Respiratory rate (breaths per min) [*n* = 173]	22 (20–28)	22 (20–24)	24 (20–32)	0.018
Systolic blood pressure (mm Hg) [*n* = 303]	110 (110–120)	110 (110–120)	110 (100–120)	0.066
Diastolic blood pressure (mm Hg) [*n* = 303]	70 (70–80)	70 (70–80)	70 (70–80)	0.026

Data are *n* (%), median (IQR); * includes referred, LAMA or unknown; ‡ coronary artery disease or congestive heart failure.

**Table 3 healthcare-12-01691-t003:** Median recovery time of COVID-19 patients by socio-demographic characteristics and patient’s condition at admission (*n* = 507).

Variables	Number	Median Recovery Time	Log Rank χ2-Value	*p*-Value
		Point Estimate (95% CI)		
Age group, years				
<20	10 (2.0)	9 (6.63–11.36)	7.11	0.212
20–29	30 (5.9)	9 (6.31–11.68)		
30–39	66 (13.0)	9 (7.25–10.74)		
40–49	104 (20.5)	8 (6.88–9.11)		
50–59	127 (25.0)	9 (7.68–10.31)		
60–69	170 (33.5)	12 (10.16–13.83)		
Sex				
Male	345 (68.0)	9 (8.11–9.88)	0.004	0.947
Female	162 (32.0)	9 (7.07–10.92)		
Origin of Residence				
Dhanusha	255 (67.1)	9 (7.85–10.14)	2.60	0.626
Mahottari	76 (20.0)	10 (6.74–13.25)		
Sarlahi	26 (6.8)	10		
Siraha	17 (4.5)	18 (10.16–13.83)		
Bara/Parsa/Rautahat/Saptari	6 (1.6)	9 (8.03–9.96)		
Missing	127			
Area of Residence				
Urban	299 (80.4)	10 (8.89–11.10)	0.005	0.945
Rural	73 (19.6)	9 (7.23–10.76)		
Missing	135			
Types of Hospital				
Public	285 (56.2)	9 (8.05–9.95)	6.60	0.010
Private	222 (43.8)	10 (8.78–11.21)		
Severity at admission				
Mild	103 (22.3)	7 (5.18–8.81)	39.42	<0.0001
Moderate	157 (34.0)	9 (7.78–10.22)		
Severe	135 (29.2)	10 (8.59–11.40)		
Critical	67 (14.5)	18 (11.96–24.03)		
Missing	45			
Respiratory support				
None	80 (20.0)	5 (4.08–5.91)	90.16	<0.0001
Oxygen mask	260 (64.8)	10 (8.98–11.01)		
Mechanical Ventilation	61 (15.2)	22 (9.13–34.86)		
Missing	106			

**Table 4 healthcare-12-01691-t004:** Median recovery time of COVID-19 patients by reported symptoms and pre-existing conditions at admission (*n* = 507).

Variables	Number	Median Recovery Time	Log Rank χ2-Value	*p*-Value
		Point Estimate (95% CI)		
Fever				
Presence	310	9 (7.93–10.06)	0.213	0.644
Absence	123	9 (7.65–10.34)		
Missing				
Cough				
Presence	305	9 (7.84–10.15)	0.001	0.975
Absence	122	9 (7.66–10.33)		
Missing				
Fatigue				
Presence	56	10 (7.35–12.64)	0.700	0.403
Absence	121	9 (7.95–10.05)		
Missing				
Shortness of breath				
Presence	332	10 (8.66–11.33)	0.566	0.452
Absence	120	9 (7.95–10.04)		
Missing				
Respiratory distress				
Presence	16	10 (7.21–12.78)	0447	0.504
Absence	119	9 (7.95–10.04)		
Missing				
Headache				
Presence	29	9 (7.95–10.05)	0318	0.573
Absence	121	8 (5.08–10.91)		
Missing				
Pre-existing conditions				
Diabetes mellitus				
Presence	95	11 (8.07–13.92)	5.00	0.025
Absence	93	9 (7.57–10.42)		
Missing				
Hypertension				
Presence	54	11 (8.13–13.86)	0.137	0.712
Absence	103	9 (7.44–10.55)		
Missing				
Chronic obstructive pulmonary disease				
Presence	14	12 (10.85–13.14)	1.81	0.178
Absence	117	9 (7.66–10.33)		
Missing				
Asthma				
Presence	4	9 (7.95–10.05)	0.105	0.746
Absence	212	9		
Missing				
Chronic cardiac disease (excluding hypertension)				
Presence	6	7 (0.01–15.58)	0.429	0.512
Absence	118	9 (7.95–10.04)		
Missing				
Tuberculosis				
Presence	4	5 (7.69–10.30)	0.175	0.676
Absence	119	9		
Missing				
HIV/AIDS				
Presence	1	-	-	-
Absence	121	-		
Missing				
Thyroid				
Presence	16	8 (3.69–12.31)	1.69	0.193
Absence	112	9 (7.61–10.38)		
Missing				
Chronic kidney disease of any stage				
Presence	8	11	0.075	0.784
Absence	120	9 (7.63–10.06)		

**Table 5 healthcare-12-01691-t005:** Predictors of COVID-19 recovery time among patients at designated hospitals, Madhesh, Province, Nepal, by Cox proportional hazard regression analysis.

Variables	Univariable HR (95% CI)		Multivariable HR (95% CI)			
			Model-I		Model-II	
	CHR (95% CI)	*p*-Value	AHR (95% CI)	*p*-Value	AHR (95% CI)	*p*-Value
Age (per 10-year increase)	0.90 (0.83–0.98)	0.023	0.87 (0.78–0.96)	0.006	0.88 (0.75–1.04)	0.887
Types of Hospital						
Private	Reference	-	Reference	-	Reference	-
Public	1.37 (1.06–1.77)	0.014	1.05 (0.77–1.44)	0.717	3.01 (0.30–29.86)	0.345
Severity at admission						
Mild	Reference	-	Reference	-	Reference	-
Moderate	0.70 (0.51–0.97)	0.032	0.54 (0.37–0.80)	0.002	0.62 (0.23–1.67)	0.352
Severe/critical	0.37 (0.26–0.52)	<0.0001	0.46 (0.29–0.71)	0.001	0.34 (0.15–0.79)	0.012
Respiratory support						
None	Reference	-	Reference	-	Reference	-
Oxygen mask	0.30 (0.22–0.41)	<0.0001	0.34 (0.24–0.48)	<0.0001	0.76 (0.35–1.63)	0.481
Mechanical Ventilation	0.10 (0.04–0.21)	<0.0001	0.11 (0.05–0.25)	<0.0001	0.26 (0.05–1.28)	0.098
Vital signs at hospital presentation						
Oxygen saturation (%)	1.05 (1.03–1.07)	<0.0001	-	-	1.09 (1.01–1.17)	0.018
Temperature (°F)	0.90 (0.76–1.07)	0.240	-	-	0.96 (0.71–1.29)	0.810
Heart rate (beats per min)	0.98 (0.97–0.99)	0.015	-	-	0.99 (0.97–1.01)	0.547
Respiratory rate (breaths per min)	0.94 (0.90–0.99)	0.031	-	-	1.02 (0.95–1.09)	0.536

HR < 1 indicates increased duration of hospital stay; variables entered: age, severity at admission, and respiratory support (model-I); variables entered: age, severity at admission, and respiratory support, oxygen saturation (model-II).

## Data Availability

The original contributions presented in the study are included in the article/Appendix A, further inquiries can be directed to the corresponding authors.

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
