# Peer review of "COVID-19 Recovery Time and Its Predictors among Hospitalized Patients in Designated Hospitals in the Madhesh Province of Nepal: A Multicentric Study"

_healthcare, 2024, doi:10.3390/healthcare12171691_

Round 1

Reviewer 1 Report

Comments and Suggestions for Authors

Thank you for the opportunity to review this interesting study. Although an enormous amount has been published about the Covid pandemic, it is useful to compare findings across populations and healthcare providers and there is much still to be learned. Generally this is a well conducted study, with sound methodology and I would welcome publication of this data, but I have some issues with the manuscript as submitted. I assume this study used manually extracted data from paper records. It would be helpful to clearly state this is the methods and to state whether the form used extracted all data from the records, or, as I suspect, only selected records. This is important because although this data is useful in its own right, other studies where electronic data extraction of all investigations held on computer systems has been possible have then through regression analysis shown up interesting correlations with unexpected findings such as d-dimer levels as well. It would be helpful to state in the methods, that data was extracted manually and whether the extraction method extracted all available data or only selected records. This should then be mentioned in the discussion as a potential limitation. The paper would also benefit from a clearer definition of the differences between shortness of breath and 'chest distress,' and how 'chest distress' may differ from the more usually used term of 'respiratory distress.' The font name Palatino Linotype needs removing from Figure 1. The Kaplan Meier plots would benefit from re-scanning at a higher resolution or re-drawing them.

Comments on the Quality of English Language

Generally the standard of English is very high

Author Response

REVIEWER 1 

 Comments and Suggestions for Authors

Thank you for the opportunity to review this interesting study. Although an enormous amount has been published about the Covid pandemic, it is useful to compare findings across populations and healthcare providers and there is much still to be learned. Generally this is a well conducted study, with sound methodology and I would welcome publication of this data, but I have some issues with the manuscript as submitted.

Response: Thank you very much for your comments and suggestions. We have addressed all of your comments and suggestions. All changes have been marked with blue coloured writing to allow reviewer’s verification.

 I assume this study used manually extracted data from paper records. It would be helpful to clearly state this is the methods and to state whether the form used extracted all data from the records, or, as I suspect, only selected records. This is important because although this data is useful in its own right, other studies where electronic data extraction of all investigations held on computer systems has been possible have then through regression analysis shown up interesting correlations with unexpected findings such as d-dimer levels as well. It would be helpful to state in the methods, that data was extracted manually and whether the extraction method extracted all available data or only selected records. This should then be mentioned in the discussion as a potential limitation.

Response: Thank you for pointing out this. We have revised this in the method section and added this as one of the limitations.

The paper would also benefit from a clearer definition of the differences between shortness of breath and 'chest distress,' and how 'chest distress' may differ from the more usually used term of 'respiratory distress.'

Response: We used the standard term “respiratory distress” in this revised version.

The font name Palatino Linotype needs removing from Figure 1. The Kaplan Meier plots would benefit from re-scanning at a higher resolution or re-drawing them.

Response: Corrected.

Reviewer 2 Report

Comments and Suggestions for Authors

Dear authors,

I read your manuscript with pleasure, and I hope that the comments at the bottom can help improve it:  

  • DOI: 10.3390/vaccines10101602

The introduction is fascinating and well done. However, it does not even marginally address the topic of the vaccine, the possible consequences, and the ethical aspects that have characterized the vaccination campaigns (please refer to DOI: 10.3390/vaccines10101602 ). Moreover,  the gap is intended to be filled, and the aims should be better highlighted. 

Please indicate the name of the statistician who carried out the analyses.

Figure 2 - 3 and all the tables are not self-sufficient, please provide

In general, the article was well-written and very interesting.

Kind regards.

Author Response

REVIEWER 2

 Dear authors,

I read your manuscript with pleasure, and I hope that the comments at the bottom can help improve it:  

  • DOI: 10.3390/vaccines10101602

The introduction is fascinating and well done. However, it does not even marginally address the topic of the vaccine, the possible consequences, and the ethical aspects that have characterized the vaccination campaigns (please refer to DOI: 10.3390/vaccines10101602 ). Moreover,  the gap is intended to be filled, and the aims should be better highlighted. 

Response: Thank you for these important comments. We have addressed the first comments with changes made in the introduction by citing the given interesting study. In addition, we have made modifications in aim of the study highlighting the research gaps as suggested.

Please indicate the name of the statistician who carried out the analyses.

Response: We have added, formal analysis, JKS, DA, and SG; statistical analysis, JKS and DA in the authors’ contribution section of this revised manuscript.

Figure 2 - 3 and all the tables are not self-sufficient, please provide

Response: Figure 2-3 and all the tables are revised and made sufficient as suggested.

In general, the article was well-written and very interesting.

Response: Thank you very much for your comments and suggestions. We have addressed all of your comments and suggestions. All changes have been marked with blue coloured writing to allow reviewer’s verification.

Reviewer 3 Report

Comments and Suggestions for Authors

Comments

 This study aimed to determine the COVID-19 recovery time and identify its predictors among hospitalized patients in the Dhanusha District of Madhesh Province, Nepal. The study used the Kaplan–Meier Survival Curve to estimate the recovery time from COVID-19, and Cox proportional hazard model to identify the predictors of recovery time. However, the study has notable grammatical errors and incomplete sentences that need to be attended to before the study is published. I suggest enlisting the help an English language editor. The methodology is well explained and could easily be replicated in other settings. I have a few comments as follows:

Major comments

·         The study should be clear on the proportion of males vs females, for instance was the 49.5% who recovered and the 16.8% who died only from the males or these figures also included the females?  

·         In Table 1, what do the p-values mean since they were above the 20% significance level set by the authors yet variables used in the analysis were still selected? For instance males were 68% and p-value 0.675 yet the analysis was still conducted by sex?

·         It is not clear how many models were run for this analysis and how the variables that were selected at the 20% significance level were then used in the final model.

·         Actually the study lacks clarification of the modelling process or steps involved before the production of the results tables presented in the study.

·         The authors refer to model 1 and 2 in Table 5, without stating if the model performance or validation for both models was compared in order to select the best model, for instance based on AIC or similar test statistic.

·         The study lacks clarification on the modeling steps involved and must be well explained so that the reader can be able to follow the results that were presented in the tables and understand from which model they came.

Minor comments

·         Figure 2 should also show Recovered instead

·         Line 37 is incomplete

·         Males only? What about females in line 37?

·         Line 108 is incomplete.

·         Check grammar in line 211

·         In line 213, it is ‘…opted to leave…’

·         Table 3 and 4 column ‘Number’ can be written as ‘Number of Days’ or just ‘Days

·         One would expect additional or supplementary materials on model formulation in order to guide the reader on the modelling process.

·         Also how would the authors explain the non-significant p-values when the 95% confidence interval does not include or excludes zero?

Comments on the Quality of English Language

None

Author Response

REVIEWER 3

Comments and Suggestions

 This study aimed to determine the COVID-19 recovery time and identify its predictors among hospitalized patients in the Dhanusha District of Madhesh Province, Nepal. The study used the Kaplan–Meier Survival Curve to estimate the recovery time from COVID-19, and Cox proportional hazard model to identify the predictors of recovery time. However, the study has notable grammatical errors and incomplete sentences that need to be attended to before the study is published.

Response: Thank you very much for your comments and suggestions. We have addressed all of your comments and suggestions. All changes have been marked with blue coloured writing to allow reviewer’s verification.

 I suggest enlisting the help an English language editor. The methodology is well explained and could easily be replicated in other settings.

Response: Agree. We have now checked the English and grammar throughout the manuscript and two of the academics who are expert in writing in academics (DA and DN) to make it suitable for the publication.

I have a few comments as follows:

Major comments

  • The study should be clear on the proportion of males vs females, for instance was the 49.5% who recovered and the 16.8% who died only from the males or these figures also included the females?  

Response: Corrected as suggested.

  • In Table 1, what do the p-values mean since they were above the 20% significance level set by the authors yet variables used in the analysis were still selected? For instance males were 68% and p-value 0.675 yet the analysis was still conducted by sex?

Response:  In Table 1, the second column (All patients) shows the gender distribution, with 68% male and 32% female. However, a p-value of 0.675 was calculated for the bivariate analysis between gender (male/female) and COVID-19 patient outcomes (recovered/died). All variables that were significant at a significance level of less than 20% (p < 0.20) in the univariate analysis were included in the model development.

  • It is not clear how many models were run for this analysis and how the variables that were selected at the 20% significance level were then used in the final model.

Response: Three models were developed: the first was unadjusted, the second was adjusted for age, and the third was adjusted for both age and sex. The variable selection process involved including variables that were significant at a level of less than 20% (p < 0.20) in the univariate analysis. In exploratory analysis, it is common practice to select variables with a 20% significance level for inclusion in a regression model, particularly during the initial stages of model development. Therefore, we included all variables with a significance level of p < 0.20 in the univariate analysis to avoid prematurely excluding potential predictors. This criterion (p < 0.20) allows us to identify factors that may have weaker relationships with the outcome but could still contribute significantly in the context of other variables in the regression model.

  • Actually the study lacks clarification of the modelling process or steps involved before the production of the results tables presented in the study. The authors refer to model 1 and 2 in Table 5, without stating if the model performance or validation for both models was compared in order to select the best model, for instance based on AIC or similar test statistic.  The study lacks clarification on the modeling steps involved and must be well explained so that the reader can be able to follow the results that were presented in the tables and understand from which model they came.

Response: Three models were developed in total. The first model was unadjusted. In the second model (Model I), we included patient age and hospital-level characteristics, such as hospital type, severity at admission, and the need for respiratory support. In the third model (Model II), we further incorporated patient vital signs, including oxygen saturation, temperature, heart rate, and respiratory rate, in addition to the variables from the previous model. The model-building process was conducted using a progressive stepwise method, and standard measures of goodness of fit, such as the Akaike Information Criterion (AIC) and Bayesian Information Criterion (BIC), were used to evaluate the models. This text is now incorporated under the statistical analysis section of the methods.

Minor comments

Figure 2 should also show Recovered instead

Response: Corrected. (Changed Improved to “Recovered”)

Line 37 is incomplete

Response: Corrected.

Males only? What about females in line 37?

Response: Corrected.

Line 108 is incomplete.

Response: Corrected.

Check grammar in line 211

Response: Corrected.

In line 213, it is ‘…opted to leave…’

Response: Corrected.

Table 3 and 4 column ‘Number’ can be written as ‘Number of Days’ or just ‘Days

Response: We agree that it should be number rather than days for columns as these numbers indicated the number of observations for each variable.

One would expect additional or supplementary materials on model formulation in order to guide the reader on the modelling process.

Response: Supplementary file is attached.

Also how would the authors explain the non-significant p-values when the 95% confidence interval does not include or excludes zero?

Response: To our knowledge as a statistician, If the 95% confidence interval includes zero and the p-value is non-significant, this is expected and indicates no statistically significant effect. However, if the 95% confidence interval excludes zero but the p-value is still non-significant, this suggests a possible error in calculation, reporting, or interpretation, and we should carefully review and clarify the results. This was our understanding. We would be happy to hear from you if any additional suggestions in this regard.

Round 2

Reviewer 2 Report

Comments and Suggestions for Authors

Dear authors, The paper has been significantly improved and, in my opinion, is now ready for publication. Kind regards.